



# A Submesoscale Eddy Identification Dataset in the Northwest Pacific Ocean Derived from GOCI I Chlorophyll–a Data based on Deep Learning

Yan Wang[1], Jie Yang[1, 2], Ge Chen[1, 2]

[1]School of Marine Technology, Frontiers Science Center for Deep Ocean Multispheres and Earth System, Ocean University of China, Qingdao 266100, China
[2]Laboratory for Regional Oceanography and Numerical Modeling, Laoshan Laboratory, Qingdao 266100, China

*Correspondence to*: Jie Yang (yangjie2016@ouc.edu.cn)

**Abstract.** This paper presents a dataset on the identification of submesoscale eddies, derived from high-resolution chlorophyll-a data captured by GOCI I in the Northwest Pacific Ocean. Our methodology involves a combination of digital image processing, filtering, and object detection techniques, along with specific chlorophyll–a image enhancement procedure to extract essential information about submesoscale eddies. This information includes their time, polarity, geographical coordinates of the eddy center, eddy radius, coordinates of the upper left and lower right corners of the prediction box, area of the eddy's inner ellipse, and confidence score. The dataset spans eight time intervals, ranging from 00:00 to 08:00 (UTC) daily, covering the period from April 1, 2011, to March 31, 2021. A total of 19,136 anticyclonic eddies and 93,897 cyclonic eddies were identified with a confidence minimum of 0.2. The mean radius of anticyclonic eddies is 24.44 km (range 2.5 km to 44.25 km), while that of cyclonic eddies is 12.34 km (range 1.75 km to 44 km). This unprecedented hourly resolution dataset on submesoscale eddies offers valuable insights into their distribution, morphology, and energy dissipation. It significantly contributes to our understanding of marine environments, ecosystems and the improvement of climate model predictions. The dataset is available at https://doi.org/10.5281/zenodo.7694115 (Wang and Yang, 2023).

## 1 Introduction

Submesoscale eddies (SMEs) are one of the strong ageostrophic submesoscale processes in the ocean, with horizontal scales ranging from several to tens of kilometers and vertical scales of ten to hundreds of meters. SMEs are intermediate between the mesoscale and the microscale and typically exhibit a short lifespan ranging from hours to days (McWilliams, 2019; Durand et al., 2010; Thomas et al., 2008). SMEs' spirals on the sea result from the cat's eye circulation associated with horizontal shear instability (Munk et al., 2000). SMEs are often energized by the strong mixing induced by ocean currents' instabilities, the convergence of fronts, or the influence of topographic features (Thomas, 2012; Taylor and Thompson, 2023a). SMEs are crucial in material and energy exchange, influencing biochemical cycles, marine food webs, and climate



change (Lévy et al., 2012, 2018; Wang et al., 2022b). Given their significant impact, SMEs have emerged as a prominent
area of research within oceanography.

Numerical simulations are currently the main method used to study SMEs systematically. These simulations provide
researchers with a detailed understanding of SMEs by generating a large amount of data that can be analysed to understand
their characteristics, formation, and evolution (Zhang et al., 2020; Cao et al., 2021; Dong et al., 2020; Marchesiello et al.,
2011; Chrysagi et al., 2021). However, it is essential to acknowledge that numerical simulations often idealize various
parameters in fluid mechanics, which may deviate from the ocean's complex and constantly changing nature (Garabato et al.,
2022). In contrast, other analytical methods such as satellite observations, in–situ measurements and laboratory experiments
are still insufficient for studying SMEs. Studying submesoscale processes presents two primary challenges. Firstly, these
processes operate at very small spatial and temporal scales, challenging direct field observations. Presently, the available
field observation schemes are expensive and sparse, leading to a lack of comprehensive and systematic global results (e.g.,
dense submerged buoy arrays, ship-based towed CTD measurements, etc.). Secondly, there is still a lack of a clear definition
of submesoscale processes in terms of dynamics. It appears that these processes at least include frontal instability processes
at the edges of mesoscale eddies, inertial gravity waves falling into submesoscale spatiotemporal scales, Vortex Rossby
Wave on mesoscale eddies, and SMEs, etc. (Zhang and Qiu, 2018).

Many studies have utilized machine learning methods to detect, track, and predict mesoscale eddies, owing to the abundance
of reliable altimeter observations and the well-developed theory surrounding them(Duo et al., 2019; Choi and Kim, 2018;
Franz et al., 2018; Ge et al., 2023; Huang et al., 2022). However, theoretical investigations of SMEs face a shortage of
observational data due to the inadequacy of altimeter spatial and temporal resolutions for their detection. Moreover, even
with alternative high-resolution observational approaches, submesoscale processes often remain obscured amid the large-
scale ocean processes.

Observations of SMEs have been conducted using SAR images to identify "black" and "white" eddies (Dokken and Wahl,
1996; Fu and Ferrari, 2008; Xu et al., 2015; Ji et al., 2021; Hamze-Ziabari et al., 2022). Additionally, the existence of
submesoscale processes affecting the movement of phytoplankton patches was first observed in 1980 (Gower et al., 1980).
Various methods, such as manual labelling, algorithmic identification, and machine learning, are employed to observe SMEs
(Park et al., 2012; Ni et al., 2021; Xia et al., 2022). Certain methods, such as SAR and altimeter, solely offer physical
insights into the ocean's surface and do not encompass the biological or chemical processes within the eddies. In contrast,
using phytoplankton to identify eddies enables researchers to obtain information about the composition and activity of the
biological communities residing within the eddies. It's essential to recognize that the utilization of SAR images typically
necessitates supplementary data processing and intricate algorithms for the precise identification of SMEs, which can be a
laborious and time-consuming task.

We used a combination of digital image processing, filtering, artificial intelligence, and small object detection techniques to
identify a large number of SMEs from high–resolution chlorophyll distribution images. We calculated their relevant
characteristic information to form the SMEs dataset. The paper is organized as follows: Section 2.1 provides a detailed

description of the chlorophyll data used in the study. Next, Section 2.2.1 describes the methodology used to highlight SMEs in chlorophyll images. This is followed by Sections 2.2.2 and 2.2.3, where we elaborate on the machine learning recognition

process. Finally, in Sections 3, 4, and 5, we present the results of our study, provide information on the acquisition of the dataset, and summarize the whole research.

## 2 Data and methods

### 2.1 Chlorophyll–a data

The chlorophyll–a (CHL) data used in this study were obtained by applying the OCI empirical algorithm to Level-2 data

acquired by the Geostationary Ocean Color Imager I (GOCI) aboard the Oceanography and Meteorology Satellite (COMS) (https://doi.org/10.5067/COMS/GOCI/L2/OC/2014)(Ryu et al., 2012; Hu et al., 2012). The GOCI data have a spatial resolution of 500 meters and a temporal resolution of one hour. Measurements were taken within an area of 2500 km × 2500 km (Center: 130° E, 36° N) and a 20–minute window between 0 UTC and 8 UTC from 1 April 2011 to 31 March 2021. The array size of the data is 5685 in the meridional direction and 5567 in the zonal direction. One unique feature of the GOCI is

its geostationary orbit, which allows it to continuously observe the same region of the Earth without moving relative to the ground. This makes it particularly useful for monitoring dynamic ocean phenomena such as coastal currents and ocean color. The GOCI coverage area is illustrated in Fig. 1.

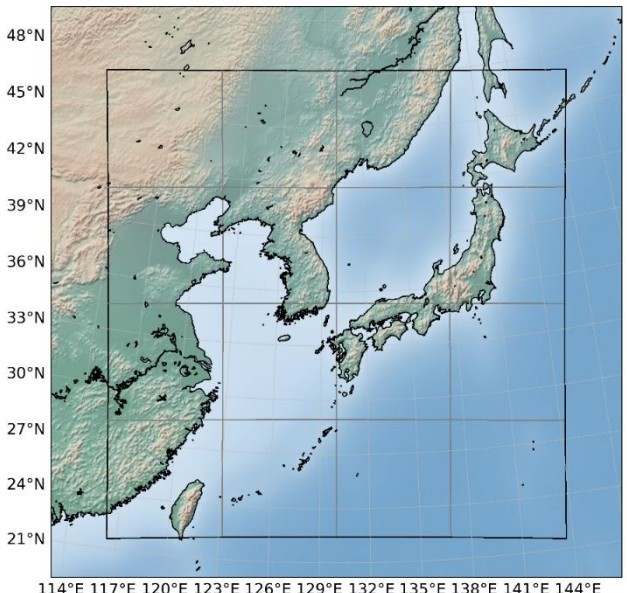

**Figure 1: The coverage area of Geostationary Ocean Color Imager I. The coverage area consists of 4 × 4 slots that overlap with**
**each other (Lambert Azimuthal Equal Area Projection).**



## 2.2 Identification method

### 2.2.1 Enhancement of chlorophyll image

Fig. 2 presents the flowchart of the CHL image enhancement technique. In the following sections, we will provide a detailed description of each step, explaining the role and parameter selection.

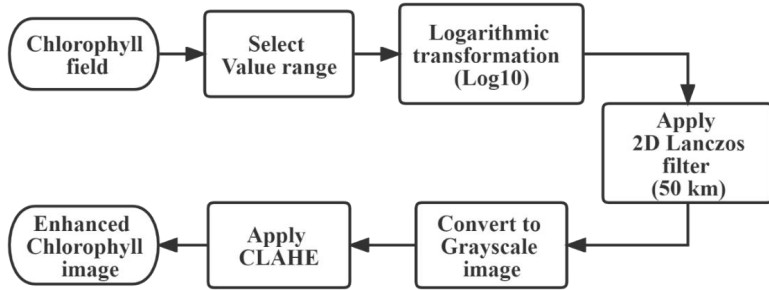


**Figure 2: The flowchart of the CHL image enhancement. We have selected a data range of 0.001–20 (mg/m3) based on the CHL value range of the satellite coverage area. Before applying the contrast-limited adaptive histogram equalization (CLAHE), we will assign a value of 0, i.e., black display, to invalid regions such as those occluded by clouds.**

In coastal areas, the vast difference in CHL concentration between coastal and oceanic regions, spanning several orders of magnitude, allows for a more straightforward visual interpretation of SMEs. However, the distinction is nearly indistinguishable in regions characterized by low CHL concentration. Therefore, applying a logarithmic transformation to CHL data is often necessary when plotting CHL fields to avoid color-stacking displays. Employing logarithmic scaling facilitates differentiation among areas with varying CHL concentrations, resulting in more lucid and informative CHL maps. Nonetheless, relying solely on logarithmic transformation proves insufficient, as shown in Fig. 3a. Large-scale circulation, mesoscale eddies, waves, and other processes at larger scales mask the CHL variability caused by submesoscale processes. A 2D Lanczos filter with a half-power cut-off wavelength of 50 km was utilized to address this issue. This choice of cut-off wavelength aligns with the sea surface height field as depicted in Fig. 3b (Pegliasco et al., 2022).

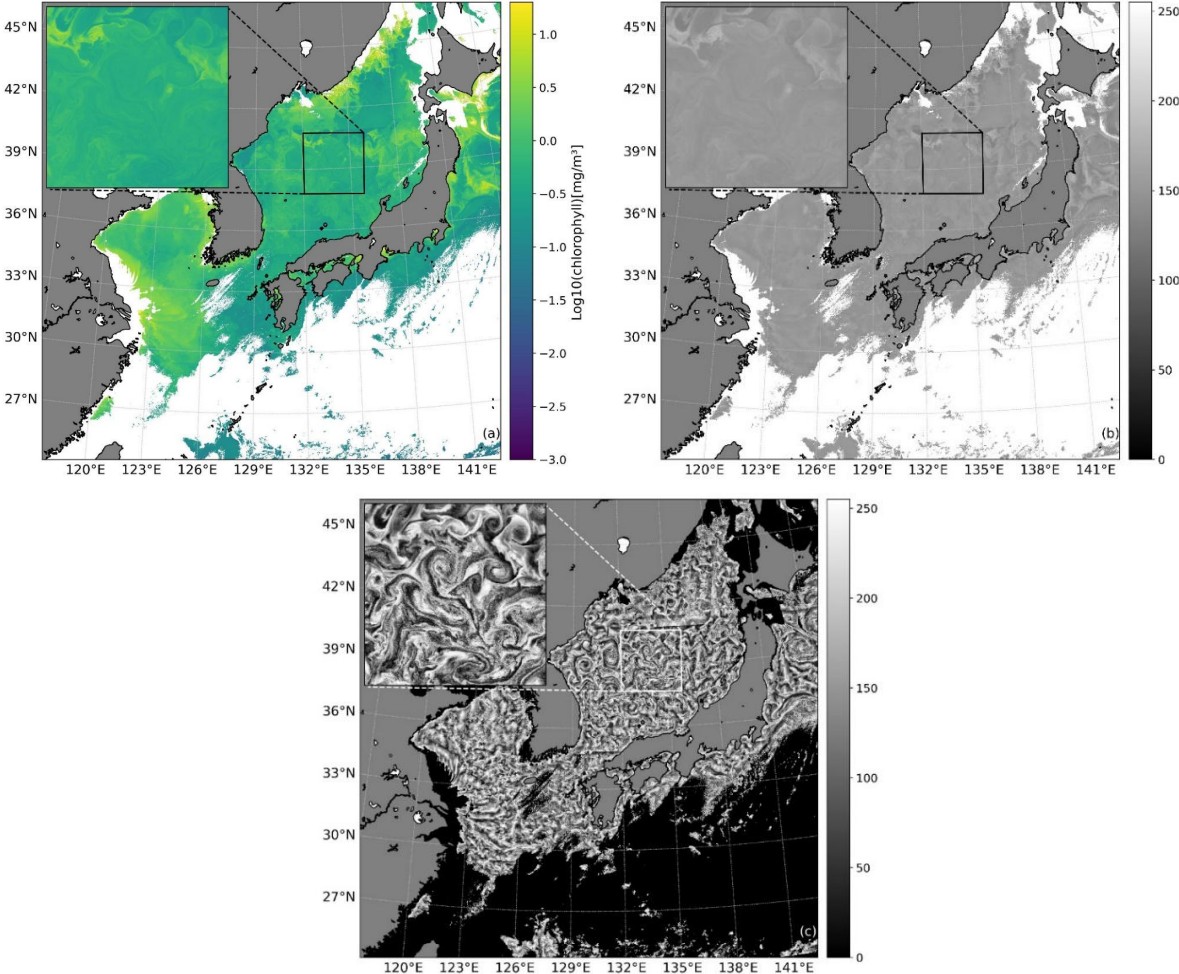

**Figure 3: A Comparison of different CHL image enhancement methods (April 5th, 2011 at 3 AM UTC). (a): The CHL values were selected in the range of 0-20 mg/m³ and then log10 transformation was applied to them. (b): Apply a 2D Lanczos filter with a half power cut–off wavelength of 50 km to image (a) and convert it to a grayscale image with a range of 0-255 for visualization. (c): The final effect of enhancing the entire CHL image by applying CLAHE to image (b). For each image, the top-left subfigure displays a 3x magnified view of the boxed region in the image, allowing for a clear visualization of the effect of each step in the image enhancement process on the high-resolution image.**

We conducted testing on the half–power cut–off wavelength of the filter and observed that when the wavelength is overly long, it tends to obscure the spiral structure within mesoscale eddies, making it challenging to distinguish SMEs and their polarity. Conversely, a too-short wavelength generates numerous discontinuous vortex filaments, making it difficult to identify relatively closed SMEs (refer to Fig. 4).

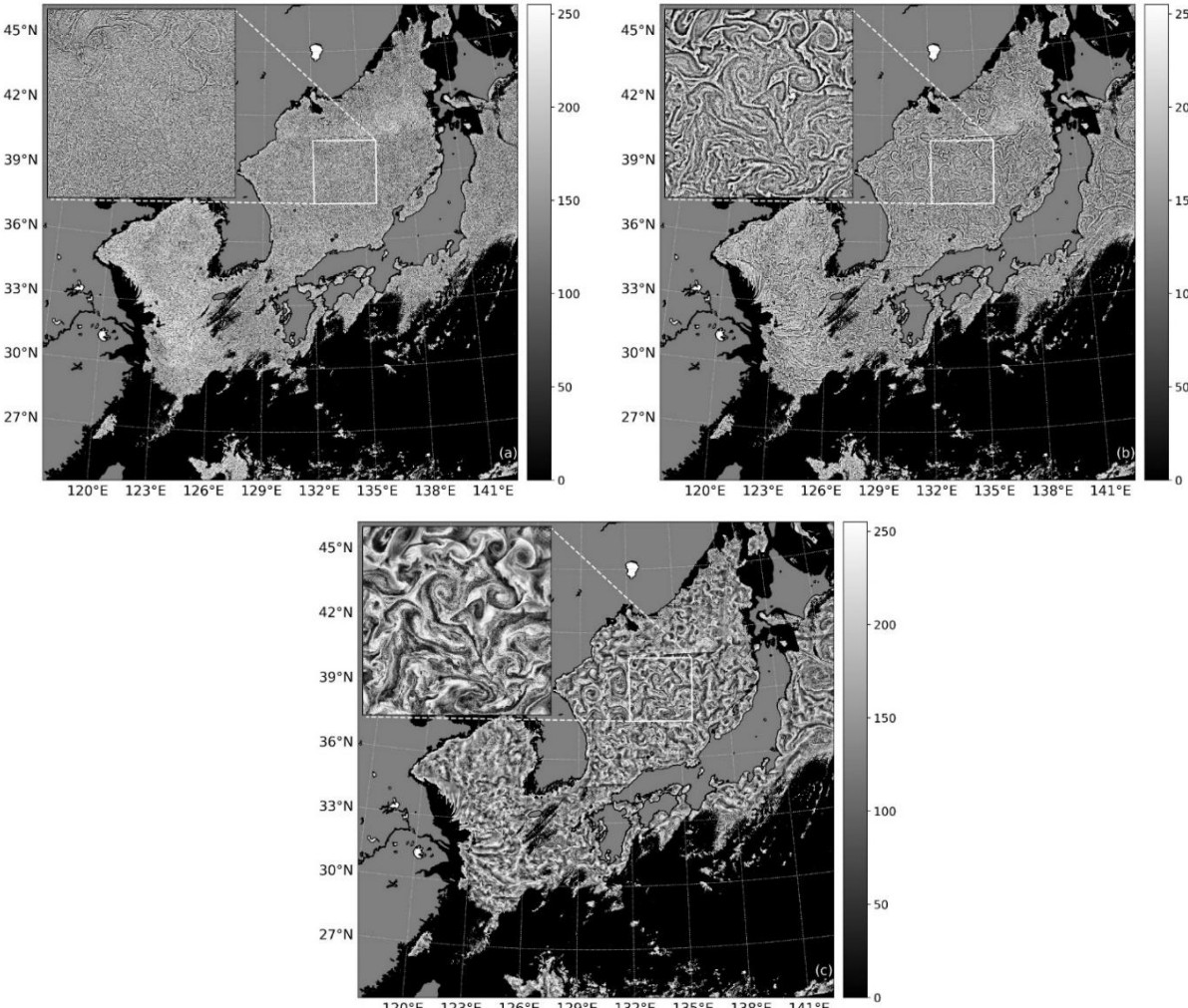

**Figure 4: A comparison of the CHL image enhancement results using different filter cut–off wavelengths (April 5th, 2011 at 3 AM UTC). (a), (b) and (c) show the results obtained by selecting a half-power cut–off filter wavelength of 1 km, 5 km, and 200 km, respectively.**

Finally, we adopted a contrast-limited adaptive histogram equalization (CLAHE) image enhancement technique to highlight the SMEs with the same display effect in the entire image (refer to Fig. 3(c)). Adaptive histogram equalization (AHE) is a widely used technique for image contrast enhancement, which calculates the image's histogram and applies a non-linear transformation to stretch the intensity values. However, AHE can lead to excessive amplification of noise in relatively uniform areas of the image. CLAHE is a modification of AHE that helps avoid this problem by limiting the amplification of the contrast to a certain predefined value (Zuiderveld, 1994; Vidhya and Ramesh, 2017). This approach involves dividing the image into small regions, called tiles, and then applying AHE to each tile individually. The CLAHE is employed to improve the clarity of chlorophyll spirals, enabling the training and identification of these spirals by AI in sea areas with chlorophyll concentration differences spanning several orders of magnitude using the same training dataset.



The general histogram equalization formula is the following Eq. (1):

$$h(v) = \text{round}\left(\frac{cdf(v) - cdf_{min}}{(M \times N) - cdf_{min}} \times (L - 1)\right) \tag{1}$$

Where $v$ represents the intensity of any pixel in the image, $h$ represents the histogram equalization function, $cdf$ is the cumulative distribution function of the image pixel intensities, $cdf_{min}$ is the minimum non-zero value of the cumulative distribution function, $M$ is the width and $N$ is the height of the image, and $L$ is the number of grey levels used (in most cases, 256).

Considering the horizontal scale of SMEs, a sliding window size of 100x100 was chosen when applying adaptive histogram equalization with contrast limiting. Furthermore, the CHL data was transformed into a grayscale image to optimize the visualization and alleviate the computational load for machine learning.

### 2.2.2 Establishment of the train set

Due to the high image resolution, it is not feasible to categorize the entire image into cyclonic, anticyclonic eddies, and non-eddy regions for eddy recognition model training. As a result, we adopted a labelling strategy that categorized labels into three types: Cyclone eddies (CE), Anticyclone eddies (AE), and bounding boxes (BOX). The discrimination between cyclones and anticyclones was based on the rotation direction of the eddy-modulated CHL spiral curves from the outside to the inside, which is consistent with the rotation direction of the two types in the Northern Hemisphere, where cyclones rotate counterclockwise and anticyclones rotate clockwise (Chelton et al., 2011; Zhang and Qiu, 2020; Wang et al., 2023). Subsequently, we extracted the BOX from high-resolution images as actual training images for the network. A total of 513 BOXs were annotated, including 160 anticyclones and 500 cyclones. To enhance model robustness and increase training sample diversity, data augmentation such as adding salt–and–pepper noise, histogram equalization, random angle rotating images, and adding random Gaussian noise to images were employed, as shown in Fig. 5.

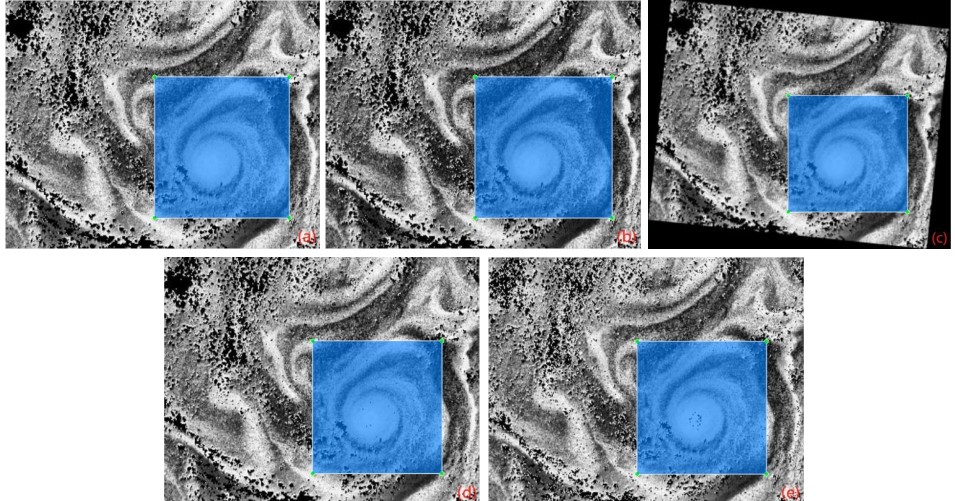



**Figure 5: Different approaches for the sample set augmentation. An example of a marked CE in a BOX, including the original image (a), images modified using histogram equalization(b), random angle rotation(c), salt-and-pepper noise(d), and random**
**Gaussian noise(e). After rotation, the parallel marking box is the minimum bounding rectangle of the rotated marking box.**

To minimize the uncertainty of establishing the training set manually, we list the following five criteria, as shown in Fig. 6.

(1)   Chlorophyll spirals should exhibit evident rotation for at least one circle.

(2)   There should be no more than a 50% overlap between adjacent SMEs.

(3)   The entire spiral structure of an eddy is supposed to be labeled rather than just its central part region.

(4)   Partially missing SMEs meeting the above three criteria should also be labeled.

(5)   When in doubt about labeling, prioritize clear eddy to annotate.

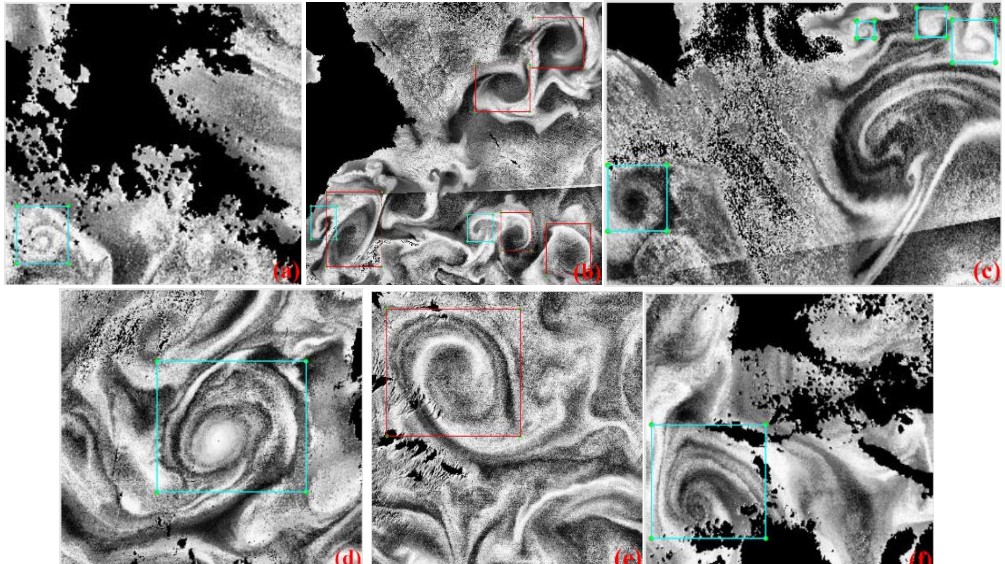

**Figure 6: Example diagram of the SMEs training set. The red box indicates AEs, and the blue box corresponds to CEs.**

### 2.2.3 Image preprocessing and SMEs identification

Detecting small targets in high-resolution images poses a highly challenging task. Small targets are characterized either by their relatively small size compared to the entire image or by having a minor difference in pixel value compared to surrounding pixels. SMEs fully comply with both definitions and are ubiquitous and intertwined with CHL fields. Therefore, we developed an image preprocessing method for identifying SMEs, which includes an image cropping method based on the eddy radius and the conversion between the image and the geographical coordinate system. The cropped image resolution is
640 * 640, and the overlap percent is calculated based on the diameter of the SMEs, following Eq. (2):

$$OP = D/(SR * PS), \tag{2}$$

Where $OP$ is the overlap percent, $D$ is the maximum diameter of SMEs (100 km), $SR$ is the spatial resolution (0.5 km), and $PS$ is the size of cropped images (640). By applying this calculation, an original image with dimensions of 5685 * 5567 can be divided into 12 * 12 small images through cropping, with each cropped image having its corresponding row and column

number in the original image. To ensure the effectiveness of the CHL data, we set a requirement that the CHL data rate in each cropped image should not be less than 5%. The geographic coordinates of the cropped image are calculated based on the row and column numbers of the cropped image and the transformation relationship between the image coordinate system and the geographic coordinate system of the original image. If $(x, y)$ is an image coordinate point in the cropped image, then its geographic coordinate $(lon, lat)$ can be calculated as follows in Eq. (3):

$$lon, lat = f\left((x + \text{col} * PS(1 - OP)), (y + \text{row} * PS(1 - OP))\right), \tag{3}$$

Where the function $f$ describes the correspondence between the original image coordinates and the geographic coordinates, and col and row represent the column and row number of their corresponding cropped images in the original image, respectively. The flowchart of the overall process of identifying SMEs and generating datasets using enhanced chlorophyll images is shown in Fig. 7.

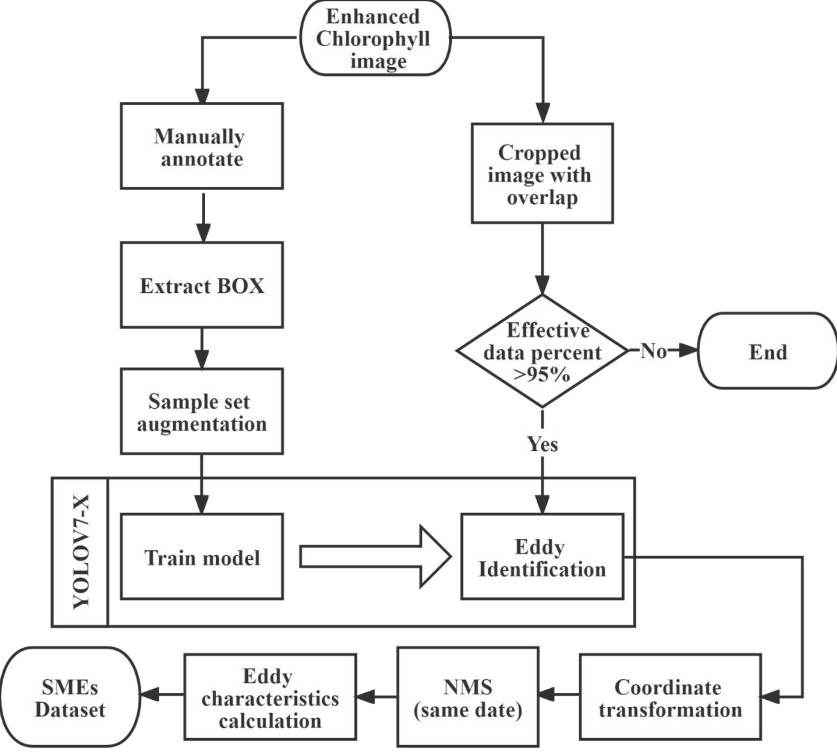


**Figure 7: The flowchart of identifying SMEs and generating SMEs datasets using enhanced chlorophyll images.**

We used the YOLOv7–X model for identifying SMEs, which perfectly balances speed and accuracy (Wang et al., 2022a). YOLOv7–X was obtained by increasing the number of layers and the number of features extracted per layer in the YOLOv7 model, aiming to amplify the model for improved performance in object detection tasks. The structure of YOLOv7–X

mainly consists of three parts: backbone feature extraction network part, strengthen feature extraction network and YOLO Head. To accelerate model convergence and reduce memory consumption, the Adam optimizer is selected to automatically



learn the parameters of all models. The loss function of our model inherits the loss function of the YOLO series, which mainly includes shape loss, confidence loss, and classification loss of the predicted box. The total loss function of object detection is defined by the following Eq. (4):

$$\text{Loss}_{total} = \text{Loss}_{shape} + \text{Loss}_{confidence} + \text{Loss}_{class}, \tag{4}$$

Where $\text{Loss}_{shape}$, $\text{Loss}_{confidence}$, $\text{Loss}_{class}$ denote the shape loss, confidence loss, and classification loss of the predicted anchor box, respectively, the confidence is a signal to judge whether the anchor box contains objects. Their basic components are binary cross-entropy loss and mean squared error loss (Redmon and Farhadi, 2018; Bochkovskiy et al., 2020; Ge et al., 2021).

Furthermore, a non-maximum suppression technique was utilized to merge them to avoid repeated identification of eddies in the overlapping regions of the cropped images. Since many eddies are formed from the same unstable currents and often overlap, we set the intersection-over-union (*IoU*) threshold for non-maximum suppression to 20%. The *IoU* is the overlap ratio between the detected box (DT) and the corresponding ground truth box (GT). The *IoU* can be calculated by the following Eq. (5):

$$IoU = \frac{S_{DT} \cap S_{GT}}{S_{DT} \cup S_{GT}}, \tag{5}$$

where $S$ represents the pixel areas of the anchor box, $S_{DT} \cap S_{GT}$ is the intersection area of $DT$ and $GT$, and $S_{DT} \cup S_{GT}$ denotes their union area.

The identification results within 5 pixels of the image edges were removed to ensure the detection of complete eddies. Within the model, flip transformation for image enhancement was turned off, and non-maximum suppression was applied to

different categories of eddies to prevent the model from classifying the same eddy differently.

**2.2.4 Cloud cover in the identification**

The results of SMEs identification based on the ocean color remote sensing images can not represent the actual distribution pattern of SMEs. The primary obstacle that affects the identification of eddies using this method is the obscuring of ocean color remote sensing signals by cloud cover, which varies across different regions, months, and times of the day. To tackle

this problem, we calculated the cloud occlusion probability (*cop*) for each *grid* using invalid CHL data, as follows Eq. (6):

$$cop(time, grid) = \frac{\sum mask(time, grid)}{fn(time)}, \tag{6}$$

Where $mask(time, grid)$ is a bool daily grid array (5685, 5567) of whether the data corresponding to hour and month are masked, and $fn(time)$ is the total number of the CHL files at the corresponding to hour and month. Therefore, by using *cop*, we can roughly calculate the number of detected eddies without cloud cover, as follows Eq. (7):

$$TN = \frac{EN}{1 - cop} \tag{7}$$

Where $TN$ is the number of eddies detected after removing cloud cover, and $EN$ is the actual number of detected eddies.



## 3 Results

### 3.1 Identification results of SMEs

We obtained 29,158 files spanning from April 1, 2011, to March 31, 2021, resulting in approximately 7.3 terabytes of data.
The chlorophyll data were extracted and utilized for image enhancement, generating a corresponding set of images. Ultimately, we obtained a total of 544,760 cropped images to identify SMEs. A total of 19,136 anticyclonic eddies and 93,897 cyclonic eddies were identified at a confidence threshold of 0.2. As shown in Fig. 8, our method can effectively identify SMEs from the chlorophyll field, and the chlorophyll spirals traced by the SMEs indicate their position and size. In the AEs, the direction of rotation of the chlorophyll spirals from the outside to the inside is clockwise, whereas in CEs, it is
the opposite. The higher the confidence in the identification results, the greater the reliability of the identification results. Using the CLAHE technique, the subtle stitching marks became visible in Fig. 8(b) and (f), which are the apparent horizontal dividing lines resulting from the joining of different slots. These stitching marks result from several minutes of measurement interval between slots, leading to variations in chlorophyll values between overlapping slots. Fig. 8(c) and (d) illustrate that the energy of the SMEs dissipates within just two hours, making it difficult to trace them in the chlorophyll field. On the
other hand, Fig. 8(e) and (f) demonstrate the effectiveness of cropped images with a 100 km overlap in preventing missed detections at the edges. Furthermore, the eddies recognized in the overlapping area differ, but they can be eliminated through non-maximum suppression.

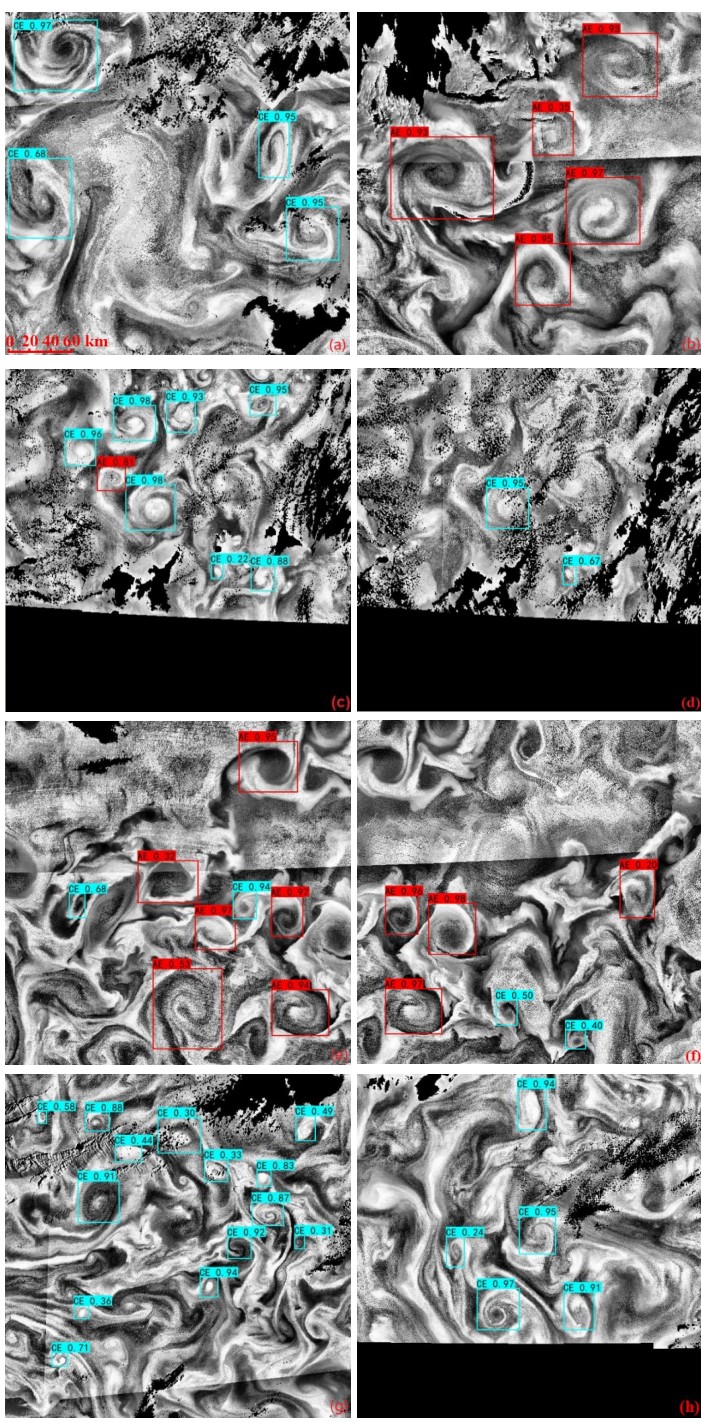

**Figure 8: Image identification results of SMEs. The blue box represents CE, the red box represents AE, and the number in the upper left corner of each identification box is the confidence score; (c) and (d) show the identification results of the same location at different times of the same day; (e) and (f) show the identification results of adjacent cropped images.**

Earth System Science Data Discussions Open Access

## 3.2 Geographic and temporal distribution of SMEs

We quantified the coverage frequency of each grid cell by AEs or CEs and reduced the correlation between the spatiotemporal distribution of SMEs and the *cop* by the method of 2.2.4. Fig. 9 shows that AEs are mainly distributed in the
Sea of Japan along the convergence zone of warm and cold currents. Conversely, CEs show a more uniform distribution, with a relatively higher concentration in the vicinity of offshore currents.

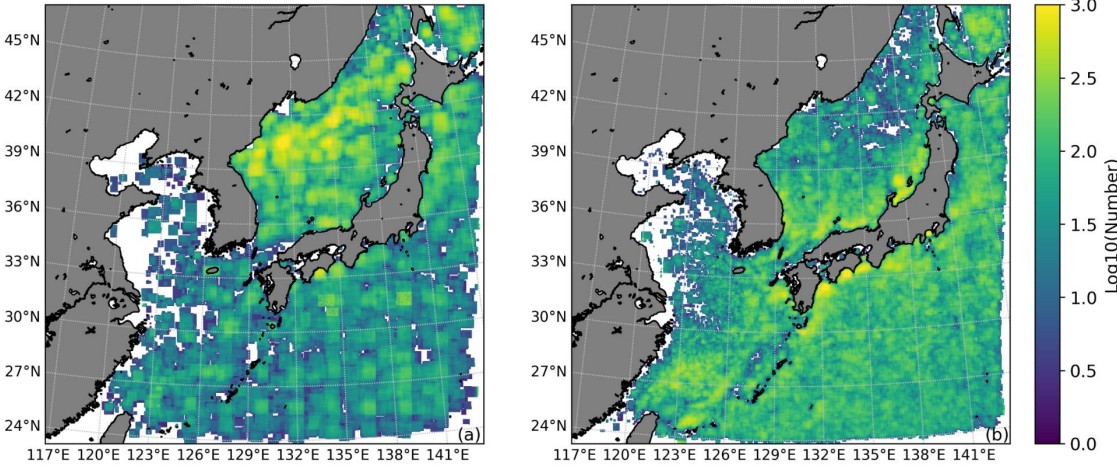

**Figure 9: Geographical distribution map of SMEs identified. The left figure is for anticyclones (a), and the right is for cyclones (b). The grids where eddies were identified were summed up by time and month after removing the cloud cover factor. Due to the**
**significant differences in the number of eddy geographical distributions, a logarithmic transformation was used for plotting.**

As shown in Fig. 10, both AEs and CEs display similar variation patterns in terms of quantity about hour and season. When calculating the local time at the central longitude of 130°E in the region, the highest number of identified SMEs occurs at around 11:40 AM. Regarding seasonal variation, both AEs and CEs experience peak numbers in April, with an additional peak in autumn. These peaks in the number of identified eddies coincide with the times of the strongest variations in sea
surface temperature, salinity, and wind conditions.

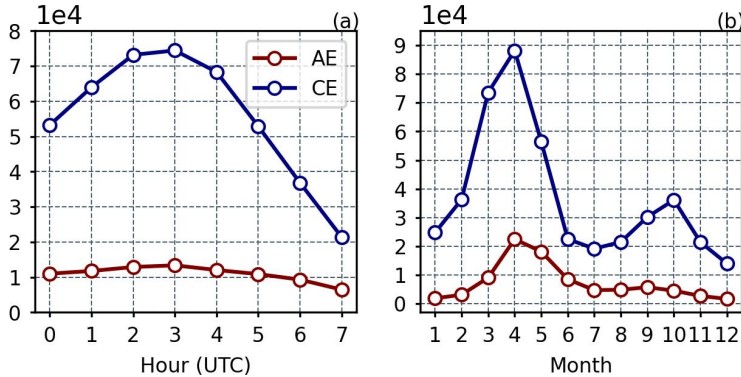

**Figure 10: Temporal variation in the number of identified SMEs. (a): The figure shows the variation in the number of identified eddies over hours. (b): The figure shows the seasonal variation in the number of identified eddies.**

### 3.3 SMEs' characteristic statistics

In Fig. 11 (a) and (b), the diameter distribution of AEs is relatively uniform, whereas the radius of CEs is concentrated within 40 km, perhaps because the CHL field stirred by smaller-scale AEs is challenging to observe. Observed AEs and CEs have the same confidence scores distribution and a majority of the detected eddies have high confidence scores in Fig. 11(c) and (d). To better study SMEs, eddies with higher confidence scores can be selected for analysis. The observed SMEs are non-geostrophic, and their diameter does not decrease with increasing latitude when comparing the estimated Rossby

deformation radius in Fig. 11(e) and (f). Additionally, it can be seen that the diameters of SMEs at different latitudes can differ by about 30 km, with the majority of CEs being smaller than the average Rossby deformation radius at the corresponding latitude.

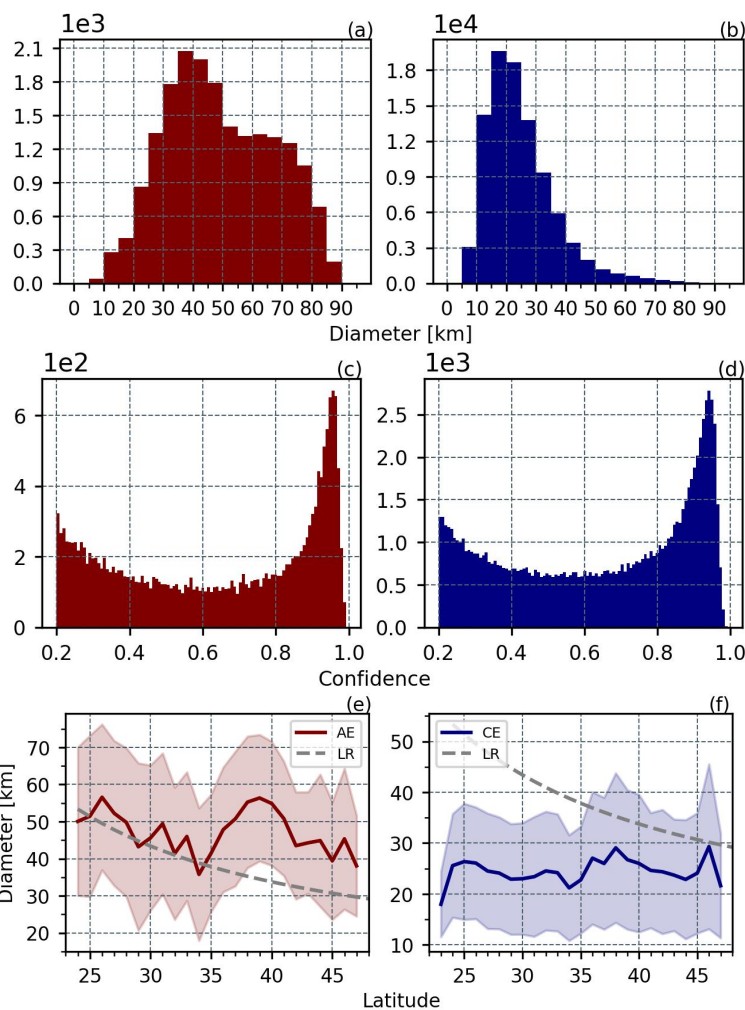



**Figure 11: (a) and (b) show the diameter distribution histograms of AE and CE, respectively; (c) and (d) show the confidence score distribution histograms of AE and CE, respectively; (e) and (f) show the variation of diameter with latitude for AE and CE and the standard deviation, respectively. Where the grey dashed line represents the variation of the Rossby radius of deformation with latitude ($L_R = \frac{(g'D)^{1/2}}{f}$), where $g'$ is the reduced gravitational acceleration, D is the water depth, and f is the Coriolis parameter.**

### 3.4 Performance of the model for eddy identification

To evaluate the detection performance of the modified YOLOv7–X, some evaluation metrics were used: precision, recall, F1-score, average precision (AP), and mean average precision (mAP). The precision and recall are defined successively using equations (8) and (9):

$$\text{Precision} = \frac{TP}{TP+FP} \qquad (8)$$
$$\text{Recall} = \frac{TP}{TP+FN} \qquad (9)$$

Where TP, FP, and FN denote the number of true positive, true negative, and false positive anchor boxes, respectively. In our experiment, the TP means the number of boxes whose $IoU$ is more significant than 0.5 between the predicted and ground truth boxes.

Besides, the F1-score measures the comprehensive performance of the network, which can be calculated based on precision and recall.

$$\text{F1-score} = \frac{2 \times \text{Precision} \times \text{Recall}}{\text{Precision} + \text{Recall}} \qquad (10)$$

The precision and recall of a specific category are used to draw curves in the 2-D coordinate system, and the area under the curve is AP of this category.

$$AP = \int_0^1 P(R)dR \qquad (11)$$

According to equation (11), mAP can be furnished, which represents the average of all categories of AP:

$$mAP = \frac{\sum_{i=1}^n AP_i}{n} \qquad (12)$$

The AP and mAP are commonly considered indicators of model quality. Generally speaking, the two indicators and model quality are positively correlated.

The evaluation metrics in Table 1 demonstrate that the modified YOLOv7–X model, trained using our method on processed and labelled samples, has achieved outstanding performance. From the recall, it can be observed that fewer AEs were identified compared to CEs, although this could be attributed to a bias in the number of training sets for AEs and CEs.

**Table 1: Precision, Recall, F1–score and AP for different categories and Mean Average Precision at $IoU$=0.5.**

|       | Precision | Recall | F1-score | AP | mAP@0.5 |
|-------|-----------|--------|----------|--------|---------|
| AE    | 100.00%   | 90.67% | 0.95     | 96.20% | 97.32%  |
| CE    | 97.80%    | 96.52% | 0.97     | 98.44% |         |

## 3.5 Validation and confidence threshold verification using Drifter hourly trajectory data

We have matched SMEs with Drifter hourly trajectory data, successfully identifying 2177 eddies whose diameters are less than 100 km, and their confidence scores range from 0 to 1. As depicted in Fig. 12(a), SMEs are primarily distributed at the periphery of mesoscale eddies, while the drifter trajectories are situated within the mesoscale eddies. To determine whether the Drifter trajectory is influenced by mesoscale eddies or is instead the result of SMEs, we refer to Fig. 12(b). In this instance, no mesoscale eddies are observed in the vicinity of the Drifter trajectory. This observation provides strong evidence that the structure of SMEs exists and possesses the capability to alter Drifter trajectories.

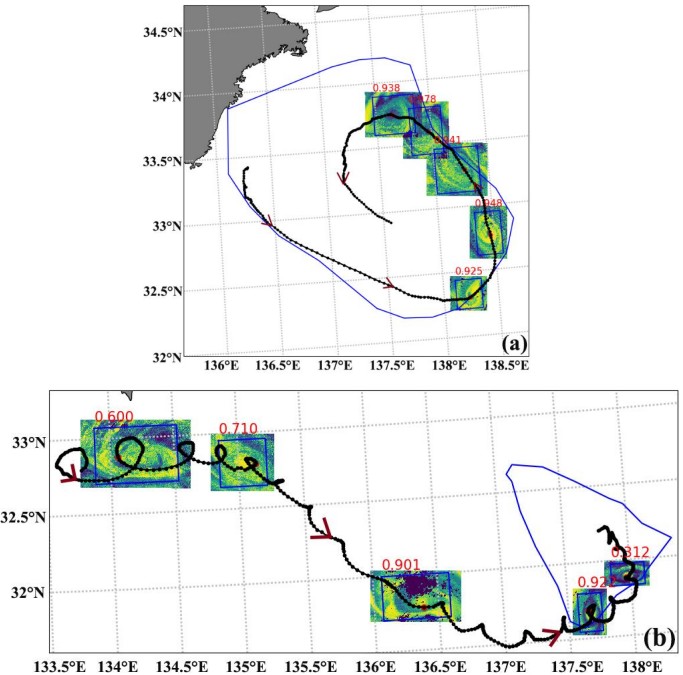

**Figure 12: Drifter trajectory and SMEs matching diagram. The black line represents the Drifter trajectory, the black points indicate hourly position points, rectangular boxes identify SMEs, and the upper numbers represent the confidence of SMEs identification. Non-rectangular shapes represent mesoscale cyclonic eddies, and red arrows indicate the drifter's movement direction. The red dots highlight the spatiotemporal matching point between the SMEs and Drifter trajectory. (a): Drifter ID 109244, with SMEs identification times from 02/05/2013 to 06/06/2013 at 00:00 daily. (b): Drifter ID 300534060522490, with SMEs identification times on 11/03/2021 at 01:00, 14/03/2021 at 02:00, 18/03/2021 at 01:00, 23/03/2021 at 04:00, and 26/03/2021 at 01:00.**

To quantitatively validate the choice of the confidence threshold, we conducted multiple experiments and discovered a correlation between the curvature variance of the Drifter trajectory matched by SMEs and the confidence level. The average diameter of this SMEs dataset is 28 km, and the typical global Drifter movement speed is 20 cm/s. Consequently, each SME can accommodate 40 drifter hourly track points. For analysis, we consider the 20 drifter hourly trajectory points before and after the spatiotemporal matching point of the eddy and Drifter trajectory. We calculate the curvature of a circle fitted to every three points, subsequently removing one maximum and one minimum to compute the curvature variance. Curvature variance can describe the degree of curvature variation of drifter



trajectories in SMEs. The smaller the curvature variance is, the more stable the influence of SMEs on Drifters and the smoother the direction change of Drifters.

As depicted in Fig. 13(a), a significant anomaly in curvature variance is observed when the confidence of eddies is below 0.2. This indicates the presence of issues, such as incorrect identification and identification of the continental margins. Fig. 13(b) reveals an inverse relationship between confidence and curvature variance. Specifically, higher confidence corresponds to smaller curvature variance, implying more pronounced and smoother chlorophyll spirals.

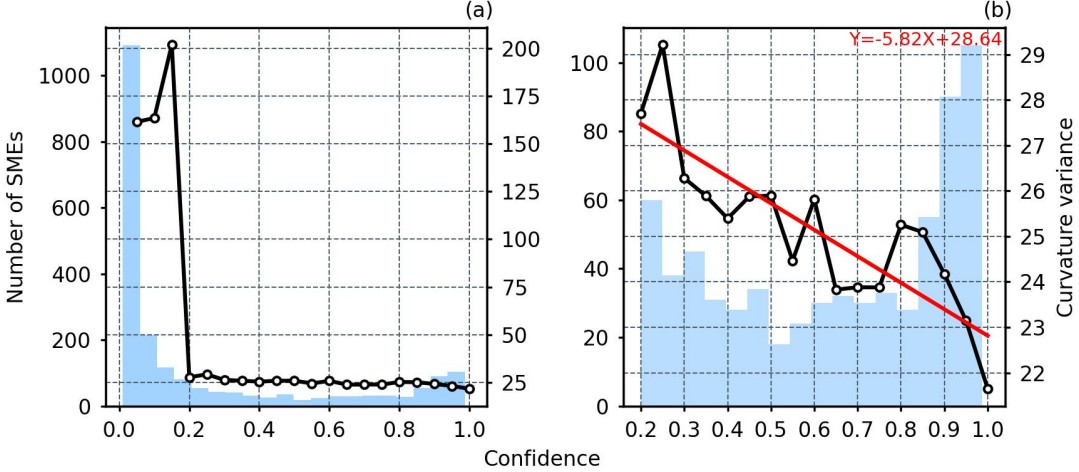

**Figure 13: Variation in curvature variance of spatiotemporal matching point between the SMEs and Drifter trajectory with confidence and the number of different confidence eddies matched.**

### 3.6 Validation and comparison using Sea Surface Temperature

We conducted a comparison between Sea Surface Temperature (SST) data and high-resolution CHL data with significant spatiotemporal trajectory overlap, as illustrated in Fig. 14. Despite the lower spatial resolution of SST data
(1km) compared to the high-resolution CHL data (500m), SMEs also conduct spiral modulation effects on SST. This means the method for identifying SMEs can be extended to sea surface skin temperature products. Fig. 14(e) and (f) reveal that eddies with higher confidence levels are more pronounced on SST. The main reason for the difference in identification results is that the deep learning model is trained according to the chlorophyll data, and the resolution of SST is 1 times worse than that of CHL.


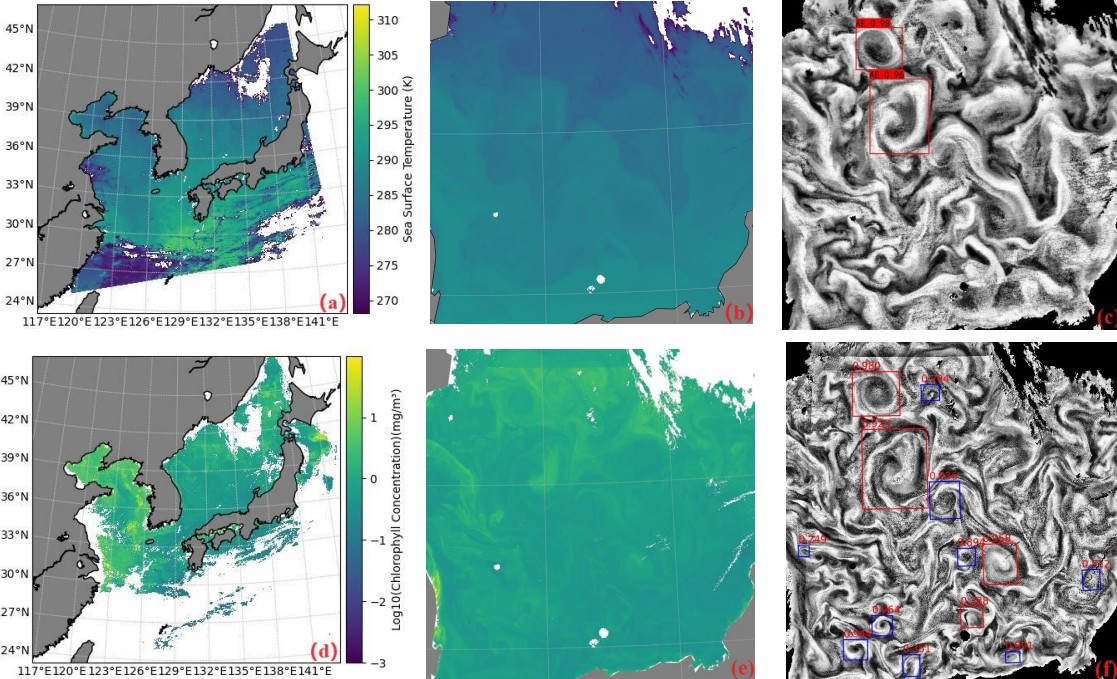

**Figure 14: A comparison of SMEs identification between sea surface temperature and GOCI chlorophyll. (a), (b) and (c) represent the MODIS Level 2 sea surface temperature at 4 o'clock on May 07, 2019 (UTC), with local amplification and image enhancement, while (d), (e), and (f) depict the corresponding chlorophyll distribution map from GOCI. The SMEs and their associated confidence have been overlaid onto the images. The red box indicates AEs, and the blue box corresponds to CEs.**

### 3.7 Validation and comparison of the identification results using Sentinel–3 chlorophyll image

The blue-green spectral bands, calculation coefficients, and image resolutions used for chlorophyll inversion are different between GOCI and OCLI sensors. Nonetheless, as indicated in Fig. 15, this method demonstrates certain applicability. Comparatively, the OCLI sensor with a resolution of 300m presents more detailed results, capable of identifying S-shaped eddies not visible in Fig. 15(c). However, due to the reliance on GOCI images for training, the confidence score of the eddy in Fig. 15(d) is diminished.

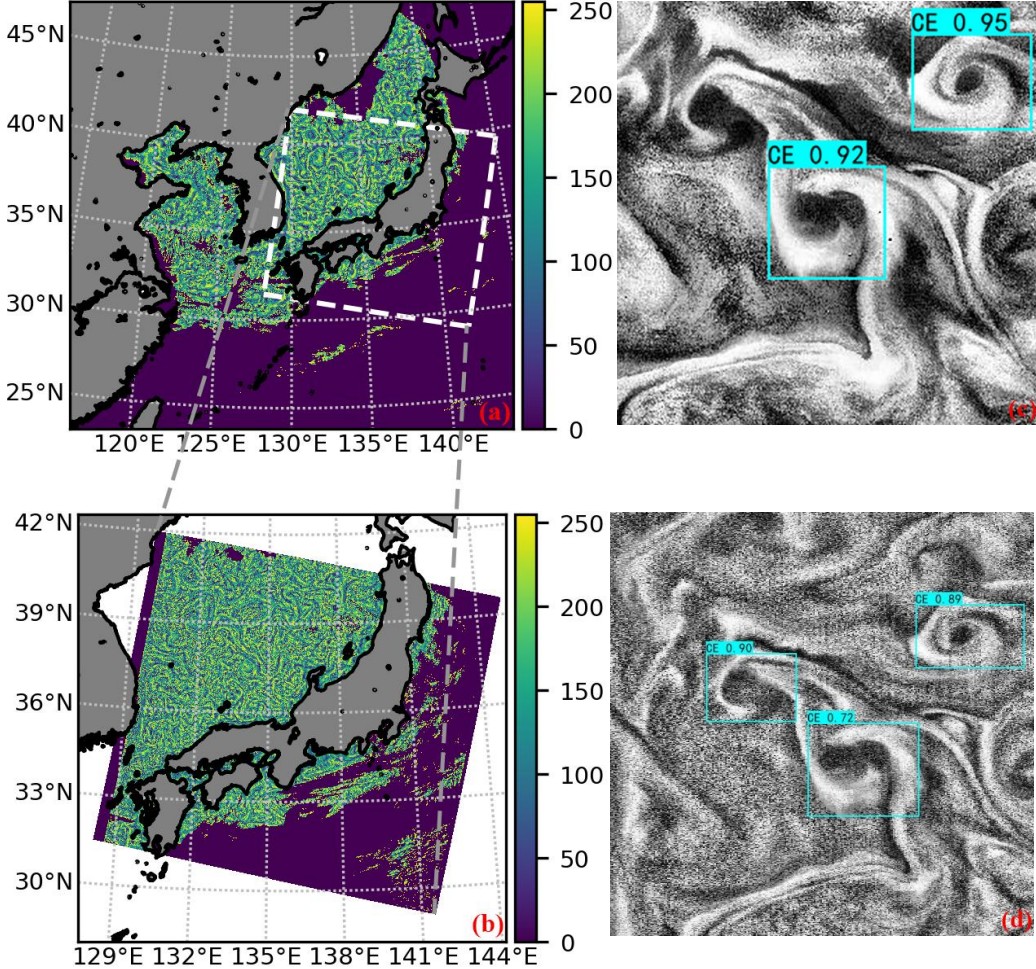

**Figure 15: A comparison of SMEs identified in chlorophyll images from the GOCI and Sentinel-3 OLCI sensors. (a) is the GOCI enhanced–chlorophyll image taken at 1:00 on May 7, 2019; (b) is the Sentinel-3B OLCI enhanced–chlorophyll image taken at a similar time. (c) and (d) are the respective identification results of (a) and (b).**

### 3.8 Validation and comparison of the identification results using the mesoscale eddy dataset

Altimetry is commonly used to identify mesoscale eddies through sea level height data. However, a global mesoscale eddy dataset is obtained by optimal interpolation, which reduces spatial and temporal resolutions. Therefore, we show the comparison between our identification results of SMEs and mesoscale eddies identified by altimetry on the same day in Fig. 16. Although altimetry identifies a more significant number of eddies and is unaffected by cloud cover, our method provides a more detailed identification of SMEs. Many eddies identified by different methods exhibit consistent spatial scales and locations. However, the altimeter fails to identify numerous SMEs within and outside the mesoscale eddies. These SMEs are reflected in our identification results, which are based on the mapping of their physical properties to the chlorophyll field.

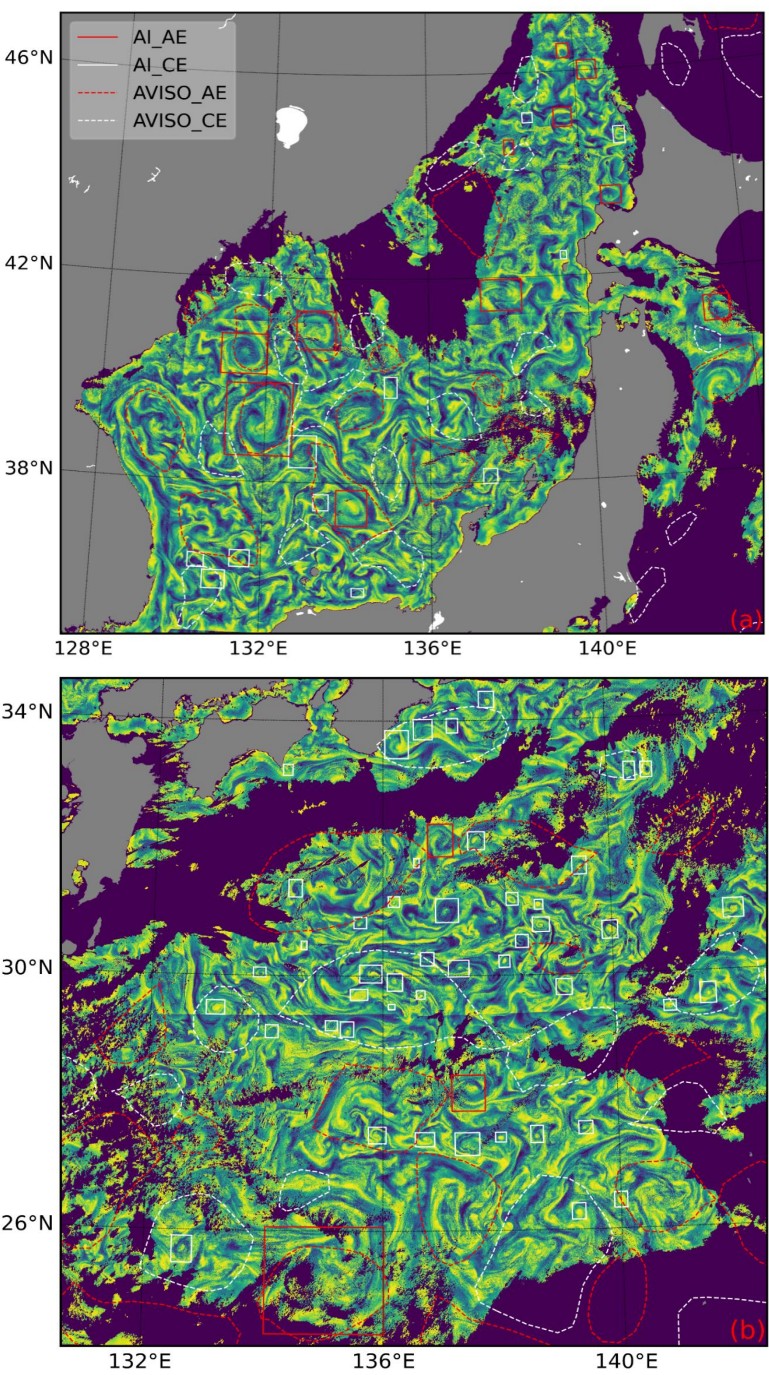

**Figure 16: A comparison between the AI eddy identification results and the AVISO eddy results on the same day with CHL-enhanced background. (a) and (b) are for May 7, 2019, and April 13, 2011, respectively.**



## 4 Datasets availability

The SMEs v1.0 dataset is saved in JSON format and can be accessed at https://doi.org/10.5281/zenodo.7694115 (Wang and Yang, 2023). The dataset contains information about each identified eddy, including polarity, location, time, geographic coordinates of the predicted box, radius of the inscribed circle, area of the inscribed ellipse, confidence score, and other relevant information.

Other data utilized in this paper can be downloaded from the following websites:

GOCI I: https://oceandata.sci.gsfc.nasa.gov/directdataaccess/Level-2/GOCI (doi: 10.5067/COMS/GOCI/L2/OC/2014)

Sentinel-3B: https://oceancolor.gsfc.nasa.gov/data/10.5067/S3B/OLCI/L2/EFR/OC/2022 (doi: 10.5067/S3B/OLCI/L2/EFR/OC/2022)

Mesoscale Eddy: https://www.aviso.altimetry.fr/en/data/products/value-added-products/global-mesoscale-eddy-trajectory-product.html (doi: 10.24400/527896/a01-2022.005.220209)

Modis SST: https://podaac.jpl.nasa.gov/dataset/MODIS_A-JPL-L2P-v2019.0 (doi: 10.5067/GHMDA-2PJ19)

Drifter hourly trajectory: https://www.aoml.noaa.gov/phod/gdp/hourly_data.php (https://doi.org/10.25921/x46c-3620)

## 5 Conclusion

Eddies can stir and maintain surface ocean chlorophyll and modulate temperature, mixing layer depth and euphotic layer depth. As a result, eddies can be observed from the chlorophyll spiral structures on the sea surface. With high spatiotemporal resolution chlorophyll data from ocean color sensors, we suppressed large-scale ocean signals by filtering and highlighted eddy-induced chlorophyll spirals by specific image enhancement. Moreover, we modified YOLOv7–X for SMEs detection and achieved a map score of 97.32% for these small targets. We identified a total of 19,136 anticyclonic eddies and 93,897 cyclonic eddies from eight CHL images per day for ten years at a confidence threshold of 0.2, with the number of cyclonic eddies being 4.9 times that of anticyclonic eddies. The mean radius of anticyclonic eddies was 24.44 km (range 2.5 km to 44.25 km), while that of cyclonic eddies was 12.34 km (range 1.75 km to 44 km). The mean radius of cyclonic eddies was half that of anticyclonic eddies. The identified cyclonic eddies were mainly concentrated in offshore flow regions, while anticyclonic eddies were primarily distributed in the Japan Sea. The number of cyclonic and anticyclonic eddies followed the same pattern over time, increasing and decreasing from around 9 a.m. to 4 p.m., with a peak around 12 p.m. There were two peaks in the seasonal variation of both types of eddies, in spring and autumn, both occurring when the mixed layer was relatively unstable. By comparing with chlorophyll products retrieved from OLCI sensors using different bands at a resolution of 300 m, we found that the modified deep learning model had a certain degree of universality. Compared with the mesoscale eddy dataset, the positions and sizes of the eddies identified by the two methods were highly similar.

However, as this is the first hour-level resolution dataset covering 10 years for SMEs in the Northwest Pacific Ocean, there are several important points to note when using it. Firstly, submesoscale activities can influence Chl-a distributions through various mechanisms, including nonlinear interactions, frontogenesis, mixed-layer instability, surface forcing, and symmetric

instability (Mahadevan, 2016). This implies that the submesoscale process is not limited to a mere form of SMEs or a spiral

structure. Secondly, differentiating between mesoscale and submesoscale motion primarily hinges on the relative significance of Earth's rotation, with the Rossby number for submesoscale motion being around 1 (Taylor and Thompson, 2023b). It's worth noting that the identification of SMEs in this paper relies on diameter, so not all of them meet the requirement that the Rossby number is approximately equal to one. Thirdly, submesoscale motions have been emphasized as potential mechanisms for transferring energy from ocean mesoscale processes to small-scale turbulence and dissipation

scales (Ferrari and Wunsch, 2009; McWilliams, 2016). In other words, the spiral structure of SMEs may not always be clear and continuous due to energy transfer. Finally, surface eddies of cyclonic vorticity are slightly more frequent than anticyclonic eddies, whereas subsurface eddies are mainly associated with anticyclonic vorticity and would be as frequent as surface anticyclonic eddies (Colas et al., 2012; Combes et al., 2015). This indicates that the SMEs dataset primarily represents surface SMEs. Furthermore, setting the confidence threshold may exclude many real SMEs to avoid retaining the

identification of disputed eddies. Nonetheless, the method proposed in this paper effectively detects SMEs, and the presence of chlorophyll spirals induced by SMEs provides a credible and direct representation of their physical properties within the chlorophyll field. These research findings hold considerable scientific significance for a deeper understanding of the role of SMEs in marine ecosystems and their impact on the marine environment.

## 6 Author contributions

Conceptualization, Y.W.; methodology, Y.W.; validation, Y.W.; visualization, Y.W.; writing—original draft preparation, Y.W.; writing—review and editing, Y.W., J.Y., K.W., M.H. and G.C.; funding acquisition, J.Y and G.C. All authors have read and agreed to the published version of the manuscript.

## 7 Competing interests

The contact author has declared that none of the authors has any competing interests.

## 8 Disclaimer


Publisher's note: Copernicus Publications remains neutral with regard to jurisdictional claims in published maps and institutional affiliations.



## 9 Financial support

This research was jointly supported by Laoshan Laboratory science and technology innovation projects
(No.LSKJ202201302), the National Natural Science Foundation of China (Grant No. 42030406, 42276203 and 42276179), the International Research Center of Big Data for Sustainable Development Goals (No. CBAS2022GSP01).

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
