# Peer review of "A Submesoscale Eddy Identification Dataset in the Northwest Pacific Ocean Derived from GOCI I Chlorophyll—a Data based on Deep Learning"

_Earth System Science Data, 2024_

## Author Comment (AC1)

Dear Reviewer,

Thank you for your detailed review and constructive feedback on our manuscript. We have studied comments carefully and have made correction.

1. Coda availability and code example. It will be beneficial if the authors could provide code for this research. Making the code available would not only enhance transparency but also facilitate further research by enabling other researchers to replicate and build upon the findings. Besides, to code examples for how to use the dataset would be highly beneficial.

**Reply:** Thank you for your suggestion. In line 376, I have added "The code is publicly available on the website below https://github.com/Asita-yan/yolov7-eddy-CHL-GOCI." Additionally, in my supporting materials document, there is code on how to read the dataset and plot the statistical histogram of the eddy radius.

| 1. import numpy as np                                                                                |
|------------------------------------------------------------------------------------------------------|
| 2. import glob                                                                                       |
| 3. import pickle                                                                                     |
| 4. import json                                                                                       |
| 5. import matplotlib.pyplot as plt                                                                   |
| 6. import matplotlib.ticker as ticker                                                                |
| 7. from tqdm import tqdm                                                                             |
| 8.                                                                                                   |
| 9. $geo_all = [np.array([]), np.array([])]$                                                          |
| 10. file_pre = 'E:\\predict\\' # The file path needs to be changed                                   |
| 11. for i in range(8):                                                                               |
| 12. $str_i = 0' + str(i) + 1/2$                                                                      |
| 13.                                                                                                  |
| 14. for month in range(12):                                                                          |
| 15. $str_month = '0' + str(month + 1)$ if month < 9 else $str(month + 1)$                            |
| 16. print(i, ' ' + str_month)                                                                        |
| 17. $geo_dis = np.zeros((2, 5685, 5567))$                                                            |
| 18. files_pre = glob.glob(file_pre + str_i + 'dataset\\????' + str_month + '??' + str_i[0:2] + '.jso |
| n')                                                                                                  |
| 19. for file in tqdm(files_pre):                                                                     |
| 20. with open(file, 'rb') as f:                                                                      |
| 21. dataset = json.load(f)                                                                           |
| 22. type_index = np.array(dataset['results']['eddy_type_AE0_CE1']) == 0                              |
| 23. a = np.array(dataset['results']['inradius'])[type_index]                                         |
| 24. b = np.array(dataset['results']['inradius'])[~type_index]                                 |
| 25. $geo_all[0] = np.concatenate([geo_all[0], a])$                                                   |
| 26. geo_all[1] = np.concatenate([geo_all[1], b])                                                     |
| 27.                                                                                                  |
| 28. fig, $(ax, ax2) = plt.subplots(1, 2, figsize=(5, 2), dpi=300)$                                   |

29. plt.subplots\_adjust(left=None, bottom=0.19, right=None, top=None, wspace=None, hspace=0. 2)

30. ax2.hist(geo\_all[1] / 1000 \* 2, bins=np.arange(0, 100, 5), color='#000080', label='CE')

31. ax.hist(geo\_all[0] / 1000 \* 2, bins=np.arange(0, 100, 5), color='#800000', label='AE')

32.

33. ax.grid(ls="--", lw=0.5, color="#4E616C")

34. ax.yaxis.set\_major\_locator(ticker.MultipleLocator(100 \* 3))

35. ax.xaxis.set\_major\_locator(ticker.MultipleLocator(10))

36. ax.xaxis.set minor locator(ticker.MultipleLocator(5))

37. ax.xaxis.set tick params(length=2, labelsize=6, which='minor')

38. ax.xaxis.set tick params(length=3, labelsize=8, which='major')

39. ax.yaxis.set tick params(length=3, labelsize=8)

40. ax.ticklabel\_format(style='sci', scilimits=(0, 1), axis='y')

41.

42. ax2.grid(ls="--", lw=0.5, color="#4E616C")

43. ax2.yaxis.set\_major\_locator(ticker.MultipleLocator(1000 \* 3))

44. ax2.xaxis.set\_major\_locator(ticker.MultipleLocator(10))

45. ax2.xaxis.set\_minor\_locator(ticker.MultipleLocator(5))

46. ax2.xaxis.set\_tick\_params(length=2, labelsize=6, which='minor')

47. ax2.xaxis.set tick params(length=3, labelsize=8, which='major')

48. ax2.yaxis.set\_tick\_params(length=3, labelsize=8)

49. ax2.ticklabel\_format(style='sci', scilimits=(0, 1), axis='y')

50.

51. fig.text(0.43, 0.03, 'Diameter [km]', fontsize=8)

52. fig.text(0.45, 0.888, '(a)', fontsize=8)

53. fig.text(0.872, 0.888, '(b)', fontsize=8)

54. plt.show()

For further modulation of submesoscale eddies on other grid data, you can refer to the analysis flow in the article titled "Euphotic Zone Depth Anomaly in Global Mesoscale Eddies by Multi-Mission Fusion Data", https://doi.org/10.3390/rs15041062. The Fig. 1 is a schematic of a method for quantifying the effects of eddies.

**Figure 1.** Schematic diagram of normalized radius calculation. Each grid represents a ZEU grid data, the grid center is approximately the geographic coordinate of the grid data, and the irregular circle is the effective boundary of the eddy with different radius. The distance between the eddy center and ZEU data point is D, and the distance between the eddy center and the eddy effective contour is R, and the azimuth angle between the eddy center and the data point is  $\theta$  (Wang et al., 2023).

**2. A discussion on the selection and optimization of these parameters would be particularly valuable, like learning rate, the number of weights.**

**R:** Thank you for your comment. Making machine learning parameters interpretable and optimizing them for the task at hand is indeed a very meaningful endeavor. In my publicly available code, I have provided comments detailing several parameter setting heuristics, such as learning rate, epochs, and weight decay. Additionally, in line 296, I added the following content:

"When training with a custom dataset, parameters can be fine-tuned based on metrics like mAP (mean Average Precision), or techniques such as learning rate schedulers can be employed to dynamically adjust the learning rate. Methods like grid search or random search can also be used to explore different combinations of weights and learning rates, with cross-validation serving as a useful tool to evaluate model performance. When selecting a learning rate, note that a higher rate can lead to instability in training and risk missing the optimal solution, while a lower rate may slow down convergence, increasing training time. Regarding the number of weights, too few may result in underfitting, whereas too many can cause overfitting. These optimizations are specialized tasks in deep learning. While technical improvements can further enhance the detection rate of submesoscale eddies, the current dataset's size and quality are already sufficient to support meaningful scientific research."

**3. How the author solves potential biases introduced by the predominance of cyclonic over anticyclonic eddies in the training dataset is not mentioned in the manuscript. In general, an imbalance in the number of data samples can significantly influence the behavior of a deep learning model.**

**R:** Thank you for found the underlying incorrect ratio for the identified of different types of eddies. In target detection tasks, no special measures are usually required if the sample size of one category is less than three times that of another. In this case, the number of CE samples is approximately 2.9 times that of AE. Therefore, we do not believe that sample imbalance has caused any identification errors. Especially if in fact there is a quantitative difference between the AE and CE, the model trained with the same proportion of samples will affect the target detection results. However, if there is indeed a quantitative difference between AE and CE in the actual data, training the model with equal sample proportions may affect the detection results.

Given that the ratio in the detection results is 4.9, this discrepancy could impact subsequent scientific research and lead to erroneous conclusions. To address this issue, I downsampled the CE category to equalize the number of AE and CE samples. After data augmentation, 674 CE and 673 AE samples were used to retrain the model. Due to the reduced training sample size, the retrained model achieved an mAP@0.5 of 81.58%. For the image data collected at 03:00, eddy detection identified 2,193 AE and 4,461 CE instances, with recall rates of 58.42% and 81.54%, respectively. This suggests that even with a model trained on balanced samples, the detection counts for the two categories still exhibit a obvious difference.

Given the inherent unpredictability and lack of transparency in deep learning models, relying solely on detection results is insufficient to fully explain these differences. It remains necessary to provide theoretical evidence to determine whether an actual imbalance exists in the occurrence of different types of submesoscale eddies. Nonetheless, since the detected eddies in the dataset were correctly identified, the data can still support meaningful scientific research.

I added it to line 229.

**Technical Corrections:**

Paragraph 15: confidence minimum of 0.2 is unclear. "minimum confidence threshold of 0.2" may be more clear.

**R:** We sincerely thank the reviewer for careful reading. As suggested by the reviewer, We changed "confidence minimum of 0.2" to "minimum confidence threshold of 0.2" throughout the text.

Thank you once again for your valuable comments and suggestions. We believe these revisions will enhance the quality of our manuscript. The revised manuscript should be uploaded shortly.

---

## Author Comment (AC4)

Dear Reviewer,

Thank you for your detailed review and constructive feedback on our manuscript. We have studied comments carefully and have made correction.

**1. In Fig.10, please amend error bar to illustrate the standard deviations in hourly , monthly and yearly eddy numbers; amend the inter-annual changes. And, accordingly, the impacts of cloud cover on the above patterns should be discussed. It will be much better if the cloud dataset is presented.**

**Reply:** Thank you for your valuable suggestions.

Figure S1 shows the variation in the average number of eddies per hour, month, and year. Panels (a), (c), and (e) are plotted using the original dataset, while panels (b), (d), and (f) are based on cloud cover–processed data. It is evident that the number of eddies is significantly underestimated in the unprocessed dataset, and since cloud cover exhibits pronounced seasonal variability, this affects the analysis of seasonal eddy patterns.

In Figures S1(a) and (b), negative values for the number of eddies can be observed. This is because the number of eddies per day does not follow a normal distribution. Therefore, for the hourly eddy number statistic, I used the total number of eddies rather than the average count for statistical analysis. It is recommended that, when analyzing the spatiotemporal patterns of submesoscale eddies using this dataset, focusing on data from a specific hour would be more appropriate. The original Fig. 10 primarily illustrates the temporal distribution of all eddies within the dataset.

[Figure]

**Figure S1: The average number of eddies per hour, month, and year. Panels (a), (c), and (e) are drawn directly from the original dataset, and panels (b), (d), and (f) are drawn from cloud-processed data.**

Cloud cover does not directly affect the number of eddies but instead leads to missed detections by obstructing satellite imaging. We have uploaded data on the probability of cloud cover for each month, hour, and grid cell (values range from 0 to 1, with higher values indicating a greater probability of cloud cover; the file is named HH_MM_cloud_probability.pkl) to the Zotero site (DOI: 10.5281/zenodo.13989785).

Thank you once again for your valuable comments and suggestions. We believe these revisions will enhance the quality of our manuscript.